# Implementation barriers and facilitators of remote monitoring, remote consultation and digital care platforms through the eyes of healthcare professionals: a review of reviews

Susan J Oudbier ![ORCID],[1,2,3,4] Sylvie P Souget-Ruff,[2] Britney S J Chen,[2] Kirsten A Ziesemer,[5] Hans J Meij ![ORCID],[1,6] Ellen M A Smets ![ORCID] [2,4,7]

For numbered affiliations see end of article.

**Correspondence to**
Susan J Oudbier;
s.oudbier@amsterdamumc.nl

## ABSTRACT

**Objectives** Digital transformation in healthcare is a necessity considering the steady increase in healthcare costs, the growing ageing population and rising number of people living with chronic diseases. The implementation of digital health technologies in patient care is a potential solution to these issues, however, some challenges remain. In order to navigate such complexities, the perceptions of healthcare professionals (HCPs) must be considered. The objective of this umbrella review is to identify key barriers and facilitators involved in digital health technology implementation, from the perspective of HCPs.

**Design** Systematic umbrella review following the Preferred Reporting Items for Systematic Reviews and Meta-Analyses statement.

**Data sources** Embase.com, PubMed and Web of Science Core Collection were searched for existing reviews dated up to 17 June 2022. Search terms included digital health technology, combined with terms related to implementation, and variations in terms encompassing HCP, such as physician, doctor and the medical discipline.

**Eligibility criteria** Quantitative and qualitative reviews evaluating digital technologies that included patient interaction were considered eligible. Three reviewers independently synthesised and assessed eligible reviews and conducted a critical appraisal.

**Data extraction and synthesis** Regarding the data collection, two reviewers independently synthesised and interpreted data on barriers and facilitators.

**Results** Thirty-three reviews met the inclusion criteria. Barriers and facilitators were categorised into four levels: (1) the organisation, (2) the HCP, (3) the patient and (4) technical aspects. The main barriers and facilitators identified were (lack of) training (n=22/33), (un)familiarity with technology (n=17/33), (loss of) communication (n=13/33) and security and confidentiality issues (n=17/33). Barriers of key importance included increased workload (n=16/33), the technology undermining aspects of professional identity (n=11/33), HCP uncertainty about patients' aptitude with the technology (n=9/33), and technical issues (n=12/33).

**Conclusions** The implementation strategy should address the key barriers highlighted by HCPs, for instance, by providing adequate training to familiarise HCPs with the technology, adapting the technology to the patient preferences and addressing technical issues. Barriers on both HCP and patient levels can be overcome by investigating the needs of the end-users. As we shift from traditional face-to-face care models towards new modes of care delivery, further research is needed to better understand the role of digital technology in the HCP-patient relationship.

## STRENGTHS AND LIMITATIONS OF THIS STUDY

⇒ We conducted a review of reviews (umbrella review) on digital health implementation, where we followed the Preferred Reporting Items for Systematic Reviews and Meta-Analyses guidelines.

⇒ This review encompasses digital health technologies which include patient interaction.

⇒ The screening and quality assessment process of this systematic review were conducted in duplicate.

⇒ A meta-analysis was not conducted since pooling of data was impossible due to the heterogeneity among studies investigating digital technologies and outcomes.

## INTRODUCTION

Globally, healthcare faces huge challenges due to an increase in individuals living with chronic diseases, an ageing population and increasing healthcare expenditures.[1] Therefore, a so-called digital transformation is advocated considering the opportunities the implementation of technology has in terms of reducing costs,[2] improving quality of care[3] and reducing the burden on healthcare professionals (HCPs).[4 5] The introduction of the electronic health record (EHR) was a major digital transformer. EHR improved overall quality of care by increasing healthcare accessibility, communication and the exchange of data between patients and

HCPs.[6–8] The introduction of digital health, defined by WHO as 'the use of information and communications technology in support of health and health-related fields', promises to be another innovation in healthcare.[9] Telemedicine, smartphone apps and wearables (ie, small electronic devices that use sensors to measure biometric values and physical activity)[10] are examples of digital health technologies that have been developed to address shortcomings in the healthcare sector. Digital health is expected to lead to improved cost-efficiency,[11] enhanced patient diagnosis[12] and improved adherence to treatment and patient outcomes.[13]

The importance of digital health is growing for several reasons. First, efficiency must be improved to address increasing healthcare costs, as well as demographic and epidemiological challenges such as increased life expectancy, increased rates of chronic diseases and population growth.[14] Second, there is huge societal demand for digital transformation in healthcare, as many processes in other sectors are now digitalised. For example, passengers in the airline industry use digital technology at every phase of their journey from booking a plane ticket, to choosing a seat and checking in for flights.[15] In contrast, digitalisation is lagging behind in the European healthcare system. This is due to difficulties in interoperating between systems, data security and privacy concerns and the uptake of technologies taking longer than anticipated.[16 17] Inadequate IT infrastructure, technical limitations and lack of a central authority for standardisation mean that the Netherlands face more challenges with the exchange of data between healthcare sectors, and international data exchanges, than other European countries.[18] A final reason for the need of digital transformation in healthcare is related to the significant data storage requirements needed to accelerate and increase the efficiency of health services via big data analytics.[7 19] In order to manage these data streams efficiently and effectively, digital transformation is a necessity.

Despite all these positive implications for healthcare, the implementation, adoption and scaling of digital health remains a challenge.[20] Many digital health technologies require some degree of patient interaction, such as the ability to communicate with the HCP within the EHR (ie, to have a video consultation or to participate in disease monitoring). Digital transformation therefore does not emerge from the enabling of technology alone; all stakeholders are required to successfully implement such technology within the healthcare system.[14] This implies that in order to guarantee adoption, it is crucial to be attentive to the usability by including the end-users of the technology (eg, the patients and HCPs), in the implementation process.[21] Even though HCPs aim to deliver high-quality care, their resistance to change is a frequently mentioned barrier to the implementation of new technologies.[22]

Thus, understanding the barriers and facilitators as perceived by HCPs is important to enable smooth introduction and efficient use of digital health technologies.[22]

Many studies and reviews have been executed on the subject of implementation in healthcare; however, research on factors that influence implementation of technology often focuses on a single technology,[23 24] or a specific disease.[25 26] Drawing from these reviews, the perspective of one of the key stakeholders within this implementation, that is the HCP, is often not the main focus of the study. To date, no study has systematically inventoried barriers and facilitators of implementation of digital health technology concentrating on the perspective of the HCP. Furthermore, no review investigated which barriers and facilitators are generic within technology implementation. That is, which barriers and facilitators apply regardless of the technology involved, and which vary per digital technology. Thus, the aim of this review is to systematically review the barriers and facilitators of implementations of digital health technologies that include patient interaction, specifically focusing on the HCP's perspective.

## METHODS
This review followed the guidelines of the Preferred Reporting Items for Systematic Reviews and Meta-Analyses (PRISMA) statement.[27]

### Eligibility criteria
For this review, we considered all eligible literature reviews (regardless of design) and meta-analyses studying the implementation of digital health technologies applying patient interaction and barriers and/or facilitators related to HCPs. Inclusion criteria were (1) the publication type being: literature review and/or meta-analysis, (2) describing barriers, facilitators, successes, inhibitions and/or failures for the implementation, adoption and maintenance of the digital health technology, (3) the technology described should encompass patient interaction and (4) the barriers and/or facilitators have been described from the perspective of the HCP (ie, medical specialists, nurse practitioners and physician assistants) (online supplemental table 1). A barrier is defined as any factor that hinders implementation of the technology. A facilitator is defined as something that enables the implementation of the technology. Implementation refers to the process of putting to use or integrating the digital health technology into the clinical workflow,[28] including the actual adoption and maintenance of the technology. The adoption of a technology is defined as the commitment or uptake of the technology by the HCPs,[29] whereas the maintenance of a technology is defined as the ability to consistently achieve the benefits of the digital technology after implementation.[28] We operationalised the perspective of the HCPs by performing data extraction in three rounds. First, we decided whether the HCP's perspective was the main focus of the included review by considering the title and research question. In that case, all barriers and facilitators were included in this review. Second, if the HCP was not a main focus of the

research, we examined whether the perspective of the HCP was mentioned as a separate stakeholder in the review. If so, we included the barriers and facilitators as reported to represent their perspective. Third, if HCPs were part of a broader group of stakeholders, including for example, patients or managers, we discussed whether the perspective of the HCP was sufficiently exposed. In case of consensus, we included the barriers and facilitators mentioned by that review. Since there is a clear distinction between care provided outside the hospital (ie, by the general practitioner, and care provided in the hospital, studies investigating the implementation/ adoption of digital technologies in primary care have not been included). Exclusion criteria consisted of (1) article written in language other than Dutch, German or English, (2) publication type such as research protocol, opinion paper, original empirical study, conference abstract, (3) no digital health technology described, (4) focus on phase other than adoption/implementation/ maintenance phase of the technology, (5) no patient interaction with the technology, (6) different target user (managerial/primary care/nurse/patient), (7) article describes a framework related to technology implementation and (8) no barriers/facilitators described.

### Information sources and search

The PICO search strategy was executed to systematically review the literature.[30] In collaboration with a medical information specialist (KAZ), reviews were identified from several bibliographic databases including PubMed, Embase.com and Clarivate Analytics/Web of Science Core Collection. The search was conducted on 29 June 2021 and updated on 17 June 2022. Search terms including synonyms, closely related words and keywords were used as index terms or free-text words: "digital*", "transform*" and "physicians". We used a methodological search filter to specify the results to reviews, meta-analyses and other summarising evidence. Duplicate articles were excluded using the R-package "ASYSD", an automated deduplication tool[31] followed by manual deduplication in Endnote (X20.0.3) by the medical information specialist (KAZ). The full search strategy used for each database is detailed in the online supplemental table 2.

### Study selection

Using the online Rayyan tool for systematic reviews, three authors independently reviewed the articles on title and abstract using the aforementioned criteria.[32] The first author (SJO) reviewed all articles, and the second and the third authors (BSJC, SPS-R) both screened 50% of the articles. Disagreements in the assessment by the reviewers were discussed until consensus was reached. A third reviewer (EMAS/HJM) was consulted when consensus could not be reached. The updated search (on 17 June 2022) was carried out by two reviewers (SJO, SPS-R), both authors screened 50% of the articles on the updated search. Articles about which there was doubt were discussed until consensus was reached. In total, four

articles could initially not be retrieved, one of which was eventually obtained via a librarian.

### Data collection and synthesis

After duplicate removal, data from the selected reviews were independently extracted by two authors (SJO, BSJC, SPS-R). Disagreements regarding the extracted barriers and facilitators were addressed in a consensus meeting. The following information was extracted: (1) first author, (2) publication date, (3) type of review, (4) digital health technologies reviewed and (5) facilitators and barriers related to implementation. A meta-analysis was not performed since pooling of data was impossible due to heterogeneity of the studies investigating digital technologies and outcomes. After extraction of the barriers and facilitators per review, three authors (SJO, BSJC, SPS-R) had a consensus meeting to classify the barriers and facilitators. Since there was overlap between the type of digital health technologies included in this review and our categorisation of the type of technologies, barriers and facilitators could be allocated to multiple categories.

We distinguished three types of digital health technologies, for which we used definitions derived from the literature. First, remote consultation (RC) is referred to as a consultation between an HCP and a patient conducted remotely via video or phone.[33] Remote monitoring (RM) is defined as 'an ambulatory, non-invasive digital technology used to capture patient data in real time and transmit health information for assessment by a health professional or for self-management' (Vegesna *et al*, p4).[34] Lastly, we considered a digital care platform (DCP) as a facilitator of digital exchange between a patient and HCP, enabling interaction, coordination or transaction.[35] To ensure the best classification possible, we assigned the digital technologies described in the reviews in duplicate (SJO, SPS-R) to the most fitting category, and in case of overlap, into multiple categories. Finally, one reviewer (SJO) independently categorised the extracted barriers and facilitators. A second reviewer (SPS-R) repeated this process of categorisation. Disagreements on barriers and facilitators were discussed until consensus was reached.

### Quality assessment

The quality of the studies was independently assessed by three reviewers (SJO, BSJC, SPS-R). BSJC and SPS-R assessed 50% of the studies, using the Joanna Briggs Institute approach for the quality for systematic reviews and research syntheses.[36] A low-quality article is defined when less than five questions within the assessment have been answered with 'Yes'. There were no exclusions made on the basis of a minimum quality threshold. Disagreements in the assessment were resolved until consensus was reached. Some questions (Q5, Q6 and Q9) from this tool did not apply to reviews other than systematic reviews (eg, scoping reviews); in these cases, we reported these questions as not applicable. We interpreted question number 8 (Q8) broadly, as to whether or not synthesis of data was in line with the research question.

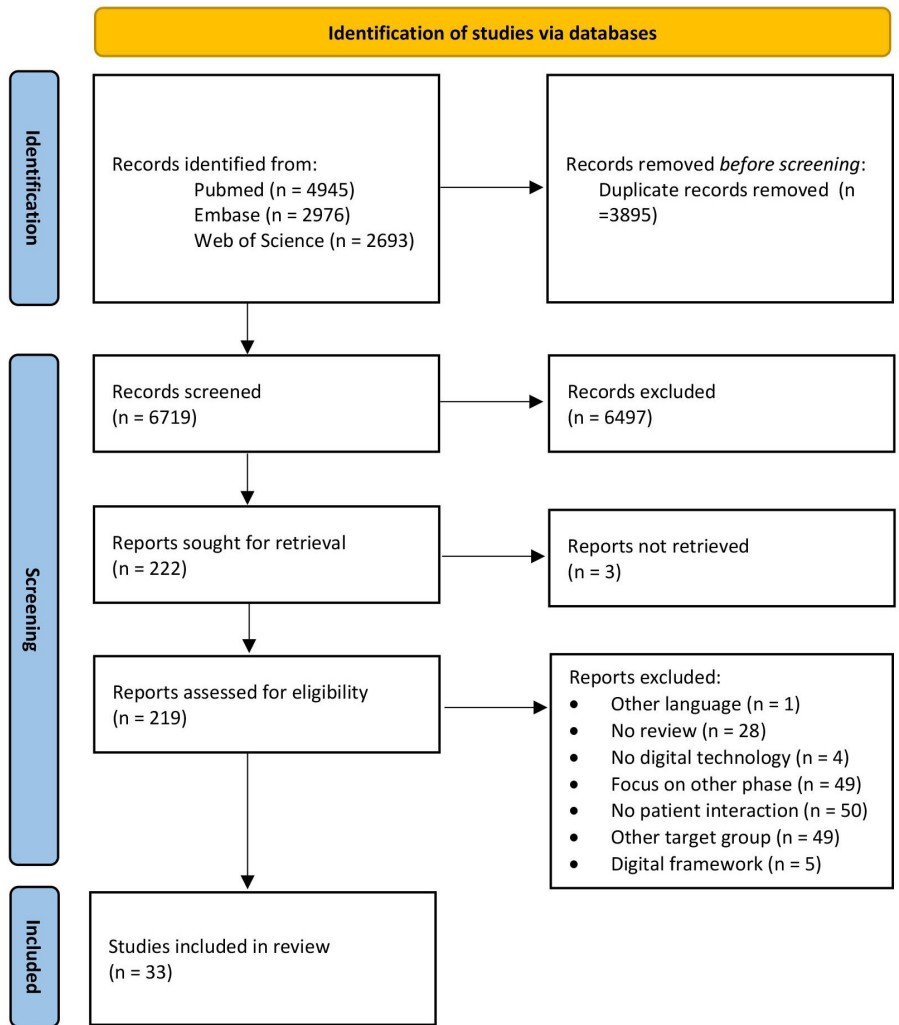

**Identification of studies via databases**

**Identification**

Records identified from:
Pubmed (n = 4945)
Embase (n = 2976)
Web of Science (n = 2693)

→ Records removed *before screening*:
Duplicate records removed (n =3895)

**Screening**

Records screened
(n = 6719)

→ Records excluded
(n = 6497)

Reports sought for retrieval
(n = 222)

→ Reports not retrieved
(n = 3)

Reports assessed for eligibility
(n = 219)

→ Reports excluded:
- Other language (n = 1)
- No review (n = 28)
- No digital technology (n = 4)
- Focus on other phase (n = 49)
- No patient interaction (n = 50)
- Other target group (n = 49)
- Digital framework (n = 5)

**Included**

Studies included in review
(n = 33)

**Figure 1** Flow chart of the article selection.

### Patient and public involvement
No patient involved.

### RESULTS
The literature search generated a total of 10 614 references: 4945 in PubMed, 2976 in Embase.com and 2693 in Web of Science Core Collection. After removing duplicates of references that were selected from more than one database, 6719 references remained. The search and screening process is illustrated in the PRISMA flow diagram in figure 1.

### Review characteristics
All reviews were published between 2012 and 2021, with more than half of the reviews (n=19) published in the last 5 years. Most reviews were either systematic (n=17) or scoping reviews (n=8). The majority of digital technologies examined in all reviews encompassed telehealth or RM of patients (table 1). Most of the diseases examined were cardiovascular disease (n=9), mental health (n=7), diabetes mellitus (n=5) and chronic obstructive pulmonary disease (n=4). Three of the included reviews reported only barriers without facilitators. The full search quality assessment is detailed in the online supplemental table 3. Two articles (Antoun[37] and Hassibian and Hassibian[38]) scored low on the quality assessment. The main reason for the reduction in quality scores was that several questions (Q5, Q6, Q9) were not applicable since they pertained to a different type of review than a systematic review.

### Barriers and facilitators reported by the included reviews
#### Generic barriers and facilitators
Barriers and facilitators from the perspective of HCPs, as identified in the included studies, have been organised into the following levels: barriers and facilitators related to the organisation, the individual HCP, the patient and the technology. Table 2 lists the generic barriers and facilitators for all digital technology types. Overall, the included reviews report less facilitators than barriers. Major overarching themes that some reviews reported as HCP-related barriers and others as facilitators include (lack of) training (n=22/33), (un)familiarity with technology (n=17/33), (loss of) communication (n=13/33), and security and confidentiality issues (n=17/33).

**Table 1** Review characteristics (n=33)

| Study | Type of technology* | Description from review article | Disease/Discipline |
|---|---|---|---|
| Almathami et al[54] | RC | Home online health consultation systems | All health conditions (mainly cardiovascular, COPD, psychotherapy, DM) |
| Antoun[37]† | RC | Electronic mail communication | Not specified |
| Appleton et al[55] | RC, DCP | Telemental health | Mental health problems |
| Batterham et al[56] | RC, DCP | E-mental health services | Depression |
| Bokolo[57]† | RC, RM, DCP | Telemedicine and virtual software | For early diagnosis and follow-up |
| Brunton et al[58] | RC, RM | Telehealth | COPD |
| Cowan et al[59]† | RC | Tele psychiatry (two-way video therapy) | Psychiatry |
| de Grood et al[60]‡ | RM, DCP | eHealth technology (EHR, EMR, telemedicine, health information technology, mHealth, e-Prescribing) | Not specified |
| De Guzman et al[61] | RC, RM | Telediabetes service | Paediatrics |
| Finkelstein et al[62]† | RC, RM, DCP | Health IT applications (telemedicine, telemonitoring, clinical decision aids, IT-guided disease management) | DM, heart disease, cancer and other |
| Gagnon et al[63] | RC, RM, DCP | mHealth applications (smartphone, personal digital assistant, remote monitoring system, etc) | Not specified |
| Greenhalgh et al[64]† | RC, RM | Telehealth (remote monitoring of symptoms, behaviour or events, biological data captured such as ECG, heart rate, blood pressure and captured data via implantable devices) | Heart failure management |
| Hassibian and Hassibian[38]† | RC, RM | Telemedicine (advice and consultation) | Chronic diseases (such as DM, hypertension, heart diseases) |
| Hopstaken et al[65] | DCP | DCPs (general information, patient-specific information, communication possibilities with their HCP via chat, e-Consult or email) | Oncology |
| Jacob et al[66] | RM | Mobile health | Mainly acute care, COPD or congestive heart failure, cardiovascular disease, DM |
| Lewinski et al[67] | RC, RM | Telehealth to perform remote triage and monitoring | Adults and their families and caregivers |
| Li et al[68] | RC, RM, DCP | eHealth applications (EHR/EMR, telemedicine/telehealth, intensive care information system, e-discharge, anaesthesia information management system, electronic logistics information system) | Not specified |
| Liyanage-Don et al[69]‡ | RM | Home monitoring | Hypertension management |
| MacLure et al[70] | RC, RM, DCP | eHealth on shared care (telemedicine, generalised eHealth implementation, EHR) | Not specified |
| Meng et al[40]‡ | RC | Teleconsultation | Neurology |
| Metzger et al[71]‡ | RC | Telemedicine | Paediatric surgery |
| Mileski et al[72] | RC, RM, DCP | Telemedicine | Hypertension |
| Muir et al[73] | RC | Online videoconferencing (psycho)therapy | Veteran mental healthcare |
| O'Cathail et al[41]‡ | RC | Patient-facing teleconsultations | Accident/Emergency, psychiatry/psychology, neurology |
| Palacholla et al[74]‡ | RC, RM, DCP | Digital health technology | Hypertension |

Continued

**Table 1**  Continued

| Study | Type of technology* | Description from review article | Disease/Discipline |
|---|---|---|---|
| Slater et al[75] | RM | Mobile health technologies | Non-communicable chronic diseases in young adults |
| Smith et al[42] | RC, RM, DCP | Telehealth | Surgery |
| Varsi et al[76] | RC, RM, DCP | eHealth programmes (mHealth, telehealth and patient portals) | Chronic conditions (arthritis, chronic pain, COPD, obesity, DM and mental disorder) |
| Vedel et al[77] | RC, RM, DCP | Health information/Telecare technologies (telecare, EHR, decision support systems, health information on the web) | Geriatrics, gerontology |
| Vyas et al[78]† | RC, RM, DCP | Telemedicine | Plastic and reconstructive surgery, dermatology |
| Whitelaw et al[79]‡ | RC, RM | Remote monitoring, tele-visits/virtual visits | Cardiovascular care |
| Xyrichis et al[80] | RM | Telemedicine | Intensive care |
| Zaman et al[81]‡ | RC, RM, DCP | Information and communication technology | Older adults with chronic diseases |

*Three types of technologies were distinguished: DCP, RC and RM.
†Other reviews than systematic.
‡Scoping reviews.
COPD, chronic obstructive pulmonary disease; DCP, digital care platform; DM, diabetes mellitus; EHR, electronic health record; EMR, electronic medical record; RC, remote consultation; RM, remote monitoring; RT, remote triage.

Generic barriers on the organisational level that were reported to be experienced by HCPs include an increased workload (n=16/33), medico-legal concerns (n=8/33) and lack of desire to change clinical paradigms (n=5/33). On the HCP level, barriers that have been reported encompass the technology possibly undermining aspects of professional identity (n=11/33), impeding the patient-clinician relationship (n=9/33) and undermining holistic surveillance (n=8/33). On the patient level, major barriers as reported by HCPs were questioning the ability of patients to use the technology (n=9/33), perceiving discomfort with engaging patients to use and to maintain to use the technology (n=4/33) and low health or general literacy of patients (n=4/33). Lastly, generic barriers reported on the technological level include technical issues such as access to Wi-Fi or unreliability of the system (n=12/33), lack of harmonisation of systems (n=10/33) and negative perception of the technology's aesthetics or design (n=3/33).

Generic facilitators on the organisational level as reported by HCPs included increased efficiency (n=13/33), scheduling capability and flexibility (n=7/33), use of champions (n=7/33) and new roles and responsibilities when using the digital technology (n=5/33). On the HCP level, the barriers and facilitators reported could act both ways, for instance, (lack of) motivation and engagement (n=11/33), level of satisfaction of the HCP (n=6/33) and that the technology could help influencing the decision-making process (n=5/33). On the patient level, patient empowerment (n=3/33) was reported by HCPs as a facilitator for implementation. Lastly, technological barriers as reported by HCPs were also reported as facilitators, and include (lack of) usefulness and HCPs perceiving no added value of the technology (n=12/33), and (lack of) usability (n=11/33).

### Barriers and facilitators divided per type of technology
Table 3 lists the barriers and facilitators divided by type of technology: RC (n=27/33), RM (n=23/33) and DCPs (n=16/33). Each type of technology is discussed below.

### Remote consultation
HCP-related barriers regarding RC include lack of feedback or evaluation of the technology, pressure from other departments or hospitals, HCPs perceiving a sense of isolation when using the technology and concerns of patients satisfaction and dependency. Facilitators regarding RC include good communication in network with other stakeholders.

### Remote monitoring
With regard to RM, barriers reported by HCPs involve loss of productivity during the implementation of the digital health technology, lack of feedback or evaluation of the digital technology, a culture that is resistant to change, difficulties establishing a trusting relationship with a patient because of distance and dependency of patients on monitoring. Facilitators specific to RM include proof of utility of the RM through research.

### Digital care platforms
Barriers reported by HCP for DCP include concerns about loss of productivity during implementation, and reduced patient satisfaction. Facilitators reported that were specific to DCP include a perceived competitive

**Table 2** Generic barriers and facilitators in digital technology implementation, from the perspective of the HCP (n=33)

| Level of barrier or facilitator | Barrier (–) facilitator (+) | # of reviews | % of reviews | References |
|---|---|---|---|---|
| **Organisational** | | | | |
| (Lack of) training | –/+ | 22/33 | | 40 54–57 59–64 66–70 72–74 76 79 80 |
| (Lack of) time and resources | –/+ | 17/33 | | 42 55 60 61 63–65 67–70 72 73 75 76 80 81 |
| Increase in workload | – | 16/33 | | 37 55 58 60 62–64 66 67 69 70 72 74 75 79 80 |
| (Lack of) leadership and support | –/+ | 15/33 | | 40 55 59–64 66–68 70 74 79 80 |
| (Lack of) reimbursement, financial incentives | –/+ | 13/33 | | 37 56 59 60 62 63 66 68 69 72 74 75 79 |
| (Lack of) integration in clinical workflow | –/+ | 13/33 | | 41 57 59 61–64 66 68 69 74 76 80 |
| Increased efficiency | + | 13/33 | | 41 42 54 55 59 62 63 65 66 70–72 79 |
| Impact on (health) outcomes | –/+ | 12/33 | | 38 41 59 63 66 67 69–72 74 80 |
| Medico-legal concerns | – | 8/33 | | 37 59 60 63 66 68 70 72 |
| Scheduling capability and flexibility | + | 7/33 | | 41 54 55 59 63 66 67 |
| (Absence of) champions | + | 7/33 | | 59–61 64 68 73 74 |
| Working relationships | –/+ | 7/33 | | 63 64 66 67 70 72 80 |
| Convenience* | –/+ | 6/33 | | 40 42 54 64 66 72 |
| New roles and responsibilities | + | 5/33 | | 66–68 76 80 |
| (Lack of) desire to change clinical paradigms | – | 5/33 | | 56 59 60 64 81 |
| (Lack of) standardisations | – | 4/33 | | 56 62 66 81 |
| Socio-political tensions | – | 3/33 | | 63 67 70 |
| Lack of purpose and implementation plan | – | 2/33 | | 63 80 |
| **Healthcare Professional** | | | | |
| (Un)familiarity with technology | –/+ | 17/33 | | 38 41 56–58 62–64 66–69 72 73 77 80 81 |
| (Loss of) communication | –/+ | 13/33 | | 37 40 54 55 63 66 67 71 72 75 78–80 |
| Undermines aspects of professional identity | – | 11/33 | | 55 58 60 62–64 66–68 70 72 |
| (Lack of) motivation and engagement | –/+ | 11/33 | | 38 54 63 64 66–68 72 73 80 81 |
| Impeding patient-clinician relationship | – | 9/33 | | 40–42 56 58 65 68 78 79 |
| Undermining holistic surveillance | – | 8/33 | | 37 40–42 54 55 58 78 |
| Fear or concerns of technology | – | 6/33 | | 37 59 60 64–66 |
| Level of satisfaction | –/+ | 6/33 | | 55 59 62 63 71 77 |
| Influence on decision-making process | –/+ | 5/33 | | 60 66 67 71 75 |
| Attitude towards technology | –/+ | 5/33 | | 38 59 63 64 73 |
| Characteristics of HCP† | –/+ | 4/33 | | 59 62 65 68 |
| (Reduce/Concerns of) overtreatment | –/+ | 3/33 | | 58 65 78 |
| Threat to face-to-face services | – | 3/33 | | 56 59 64 |
| Work-life balance | –/+ | 2/33 | | 55 67 |
| **Patient** | | | | |
| Questioning ability of patients | – | 9/33 | | 40 54 58 64–66 69 72 81 |
| Patient-tailored approach | –/+ | 4/33 | | 63 66 72 74 |
| Difficult to engage/maintain use | – | 4/33 | | 54 55 59 72 |
| Low (health/digital) literacy | – | 4/33 | | 40 66 70 72 |
| Patient empowerment | + | 3/33 | | 66 72 74 |
| Enabling access for certain groups | –/+ | 3/33 | | 55 66 75 |
| **Technical** | | | | |

**Table 2** Continued

| Level of barrier or facilitator | Barrier (–) facilitator (+) | # of reviews | % of reviews | References |
|---|---|---|---|---|
| Security and confidentiality (issues) | –/+ | 17/33 | | 37 41 60–63 65 66 68 70 72 74 75 78–81 |
| Technical issues‡ | – | 12/33 | | 40–42 62 63 66 69–71 74 75 79 |
| (Lack of) usefulness | –/+ | 12/33 | | 41 62 63 66 67 70–72 74 75 77 79 |
| (Lack of) usability | –/+ | 11/33 | | 41 54 60 62 63 65 66 69 70 72 74 |
| (Lack of) harmonisation of systems | – | 10/33 | | 60 63 65 66 68–70 72 74 79 |
| (Lack of) technical support | –/+ | 7/33 | | 40 54 66 68 74–76 |
| Perception of aesthetics/design | – | 4/33 | | 63 66 77 79 |
| Availability of data | –/+ | 4/33 | | 60 66 69 74 |

'+' indicates a facilitator, '–' indicates a barrier, '–/+' reported as barrier as well as facilitator. The barrier or facilitator is outlined in this table when reported as barrier or facilitator in all three technologies.
*Cost-efficient, travel-saving, time-saving.
†Digital literacy, age, years of practice.
‡Access to Wi-Fi, system unreliability.
HCP, healthcare professional.

advantage due to the implementation of this technology by creating new opportunities, and proven utility of the platform, preferably through research.

## DISCUSSION
This systematic umbrella review aimed to investigate the barriers and facilitators of digital health technology in patient care, as perceived by HCPs. Three types of digital health technologies were distinguished: (1) RC, (2) RM and (3) DCPs. Barriers and facilitators from the perspective of HCP could be categorised as relating to (1) the organisation, (2) the HCPs themselves, (3) the patient and (4) the technology. Overall, our review highlights that there are still major organisational requirements to

**Table 3** Barriers and facilitators of digital technology implementation from the perspective of the HCP, divided per type of technology

| | Type of technology | | |
|---|---|---|---|
| Level of barrier or facilitator | RC (n=27) | RM (n=23) | DCPs (n=16) |
| Organisational | | | |
| Loss of productivity during implementation | | (45)* | (45) |
| Lack of feedback or evaluation | (49, 52) | (49, 52) | |
| Competitive advantage | | | (53) |
| Pressure from other hospitals | (67) | (67) | |
| Good communication in network | (57) | | |
| Culture that is resistant to change | | (51) | |
| HCP | | | |
| Sense of isolation | (44, 52) | (52) | |
| Difficult to establish trust relationship | | (68) | |
| Patient | | | |
| Concerns of dependency of patients | (43) | (43, 51, 54) | |
| Patient satisfaction | (40, 44) | | (40) |
| Technological | | | |
| Proof of utility (by research) | | (45) | (45) |

The barrier or facilitator is reported when reported in less than three technologies, otherwise a barrier or facilitator is reported in the generic table (table 2).
Green background means reported as facilitator. Orange means reported as both facilitator and barrier.
*The numbers in parentheses denote the corresponding references.
DCP, digital care platform; HCP, healthcare professional; RC, remote consultation; RM, remote monitoring.

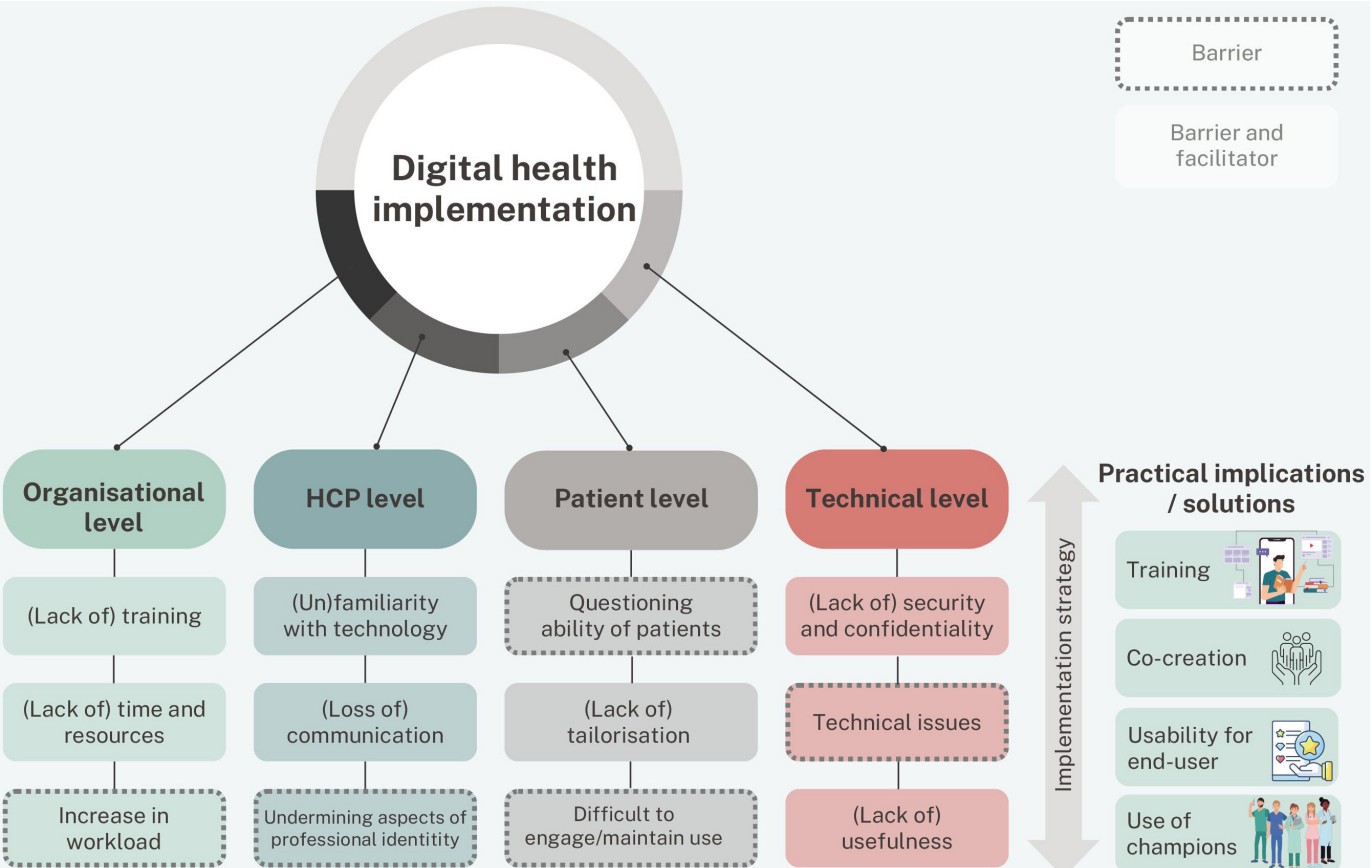

**Figure 2** Top three barriers or facilitators on each level of digital health implementation, and practical implications or solutions. HCP, healthcare professional.

be met in order to provide a solid foundation for digital health technology implementation. One way to overcome organisational barriers is by addressing them in the implementation strategy (figure 2). This includes adequate training and planning ahead for potential safety and confidentiality concerns. In addition, it also includes showing the HCP the eventual benefits such as improved health outcomes by the use of the digital health technology, and ensuring fully embedding within their clinical workflow. The results of this review indicate that the organisational facilitators as perceived by HCPs are comparable to the list of organisational barriers HCPs reported. Thus, these factors act as two sides of the same coin. Major facilitators to support a successful implementation strategy at the organisational level include deploying the right leadership and support, making use of change champions and ensuring good working relationships. However, as healthcare will continue to face new challenges with regard to technology implementation, barriers and facilitators are not fixed entities that need to be resolved for successful implementation. Instead, the implementation of digital health technologies should be approached as a dynamic process which needs ongoing management and tailoring, taking social relations and context into account.[39]

This umbrella review demonstrates that the majority of research in the past years on barriers and facilitators addressed common organisational barriers of technology

implementation, such as lack of training, lack of time and resources, an increase in workload and lack of leadership and support. This substantial attention to organisational issues can be interpreted in two ways. First, that organisational barriers might remain prevalent for implementation, hence the substantial literature on them. Alternatively, it suggests that organisational aspects could dominate discussions about implementation of technology, potentially overshadowing other levels on which barriers can take place. For instance, barriers on the level of the individual HCP, such as loss of communication, impeding the patient-physician relationship, undermining holistic surveillance or undermining aspects of professional identity, have been less frequently researched. For instance, a major barrier on the HCP level is loss of communication, specifically pertaining to RC. As a result, HCPs experienced the possibility of misunderstanding the patient, and perceived a decreased personal connection with the patient.[40 41] In addition, HCPs perceived a difference in whether one sees the patient digitally for the first time, or whether one has already established a relationship with the patient. This distinction was particularly reported in reviews implementing digital health technologies during the COVID-19 pandemic, where it became more common to have an initial digital interaction instead of face-to-face consultation.[40] In addition, technology could undermine aspects of holistic surveillance,

as physical examinations are not possible and it is more difficult to pick up non-verbal cues.[40 42] Despite the HCP perceiving these barriers, patients were found to be satisfied with teleconsultation and RM and did not experience this loss in communication.[40–42] In this regard, there is a discrepancy between the needs and perspectives of HCPs and patients. The most used explanation on differences in satisfaction levels with regard to video consultation is the inability of HCPs to physically examine the patient.[43] While this review primarily focuses on the perspective of HCP, patients are also significant end users of technology. Therefore, it is essential to consider and identify the varying needs of patients in addition to those of HCPs, by conducting more research on the patients' perspectives on the influence of digital technology on their care, and the HCP-patient relationship specifically. Such research may yield more discrepancies, which need to be solved to provide patients with best care without unduly burdening HCP.

This review also indicates the importance of training of professionals at the start of an implementation trajectory. For transformative innovations such as the introduction of a new digital health technology, adequate training to facilitate the user's adoption and use of the new technology is essential.[44] However, the training needs of HCPs and ideal frequency of training are yet to be defined. Ongoing training, rather than a one-time instruction at the start of the implementation, may be more sustainable when considering potential staff changes.[45] The training itself could vary, from practical instruction to hands-on learning, as reported in the study by Dugstad *et al*.[46] At the onset, a theoretical approach to training of professionals was used by these authors, but this was soon replaced by practical instruction and training sessions which emphasised capacity building to acquire the necessary skills; such as using the app, handling swipe hand movements, and using software commands.[46]

On the HCP level, major barriers included a loss of communication with patients, and the possibility of technology impeding the patient-HCP relationship. Patient connection is at the essence of being an HCP, and has become increasingly difficult in contemporary practice.[47] A study conducted in 50 medical departments in the Netherlands showed that HCPs with positive patient relationships were less likely to suffer from patient-related burnout even with high job demands.[47] In addition, patient-related burnout was more prevalent among HCPs who had less positive patient relationships.[48] In light of these findings, and with the increasing prevalence of burnout among HCPs,[49] further research is necessary to investigate whether the use of digital health technology in healthcare actually results in a loss of gratifying interaction with patients.

Patient-related barriers, as perceived by HCPs, relate to questioning the ability of patients to use digital technology, patient anxiety about using the technology and low patient literacy. One of the actions of WHO on their global strategy is that digital technologies should strengthen health equity by promoting an inclusive digital society.[50] As digital health literacy requires skills complementary to health literacy, it is important to ensure that digital health technologies are available for all patients.[51] Therefore, understanding and incorporating intended users' needs and context appears to be a prerequisite for effective use of digital health technology,[52] apparently also from the perspective of HCP. This could be achieved by co-creation, which is described as an interaction in which participants work together to create an outcome that is appreciated by both parties.[53] By including HCPs in this process, the possibility that the digital health technologies will meet the needs and requirements of the end user in terms of content and usability will increase; this, in turn, may enhance technology implementation.

### Strengths and limitations

The strengths of this study include its comprehensive search on digital health technologies, narrowed down to technologies that include patient interaction, its duplicate approach and its focus on the end-user. However, the current review was limited by digital technology and policies regulating these being in a state of constant flux. Therefore, barriers and facilitators reported in older studies might be less relevant over time. Presently, we acknowledge that the current state of affairs is undergoing rapid evolution, and consequently, the data of our review may not be entirely up-to-date. However, considering that half of the included reviews are <5 years old, we do not anticipate that more recent studies will have a significant impact on our findings. Another limitation is that some of the included reviews provided little context, making it difficult to determine whether findings were considered facilitators or barriers to technology implementation. In such instances, the reviewers either reached a consensus-based decision or reported a finding both as a barrier and facilitator. Finally, the included reviews differ in terms of the type of technology studied and the disease or condition in which the technology was used. The intention of this review was to provide as complete of an overview as possible of common barriers and facilitators as perceived by HCPs. All technologies in this review include a form of patient interaction, however, barriers or facilitators for a specific technology or implementation may exist.

### Conclusions and implications for practice

The implementation of digital health technology in healthcare is influenced by several major barriers from the perspective of the HCP that need to be overcome. These include security and confidentiality issues, adequate training, HCPs unfamiliarity with technology and HCPs questioning the ability of patients to use technology. In order to successfully implement digital health technology, generic barriers and facilitators as well as barriers and facilitators per type of technology must be considered. First, organisational barriers have been frequently reported and remain problematic. However, such barriers may act as facilitators when appropriately

addressed in the implementation strategy. Second, barriers on both the HCP and patient levels are related to the patient-HCP relationship, the loss of communication, the undermining of holistic surveillance and questioning the ability of patients. These barriers can be addressed by investigating the needs of the stakeholders involved and by including them as co-creators in the development of digital health technology. By involving HCPs in this process, the digital tool will be more likely to meet end-user requirements in terms of content and usability. In turn, this bolsters the chance for successful implementation. Third, successful implementation of digital health technology demands time and resources. HCP training should be a continuous process to protect against issues related to staff turnover. Finally, further research is necessary on the other end-user of digital health technology, the patient, in order to understand the role digital health technology occupies in the patient-HCP relationship, and to better understand the needs of this end-user.

**Author affiliations**
[1]Outpatient Division, Amsterdam UMC, Amsterdam, The Netherlands
[2]Department of Medical Psychology, Amsterdam UMC, Location AMC, Amsterdam, The Netherlands
[3]Amsterdam Public Health research institute, Digital Health, Amsterdam, The Netherlands
[4]Amsterdam Public Health research institute, Quality of Care, Amsterdam, The Netherlands
[5]Medical Library, Vrije Universiteit Amsterdam, Amsterdam, The Netherlands
[6]National University of Singapore Yong Loo Lin School of Medicine, Singapore
[7]Amsterdam Public Health research institute, Personalized Medicine, Amsterdam, The Netherlands

**Correction notice** This article has been corrected since it was published. Licence updated to CC BY on 2nd August 2024.

**Acknowledgements** The authors wish to thank Dr LW Peute, expert on Human Factors Engineering Methodologies and Director of the eHealth Living & Learning Lab Amsterdam (ELLLA), for her support regarding the tables and figures. Furthermore, the authors would like to express their gratitude to Rebecca Kerstens, from the University of South Australia, for critically assessing the manuscript.

**Contributors** SJO, KAZ, HJM and EMAS were involved with the design of this review. SJO, SPS-R and BSJC performed the screening of titles and abstracts of the included studies. With regard to data extraction and synthesis, this was performed by SJO, SPS-R and BSJC. SJO and SPS-R prepared the original draft of this review. HJM and EMAS contributed to the refinement. EMAS is responsible for the overall content as the guarantor. All authors have read and approved the final version.

**Funding** The authors have not declared a specific grant for this research from any funding agency in the public, commercial or not-for-profit sectors.

**Competing interests** None declared.

**Patient and public involvement** Patients and/or the public were not involved in the design, or conduct, or reporting, or dissemination plans of this research.

**Patient consent for publication** Not applicable.

**Ethics approval** Not applicable.

**Provenance and peer review** Not commissioned; externally peer reviewed.

**Data availability statement** All data relevant to the study are included in the article or uploaded as supplementary information.

**ORCID iDs**
Susan J Oudbier http://orcid.org/0000-0003-0376-3161
Hans J Meij http://orcid.org/0000-0001-8738-901X
Ellen M A Smets http://orcid.org/0000-0002-8145-8595

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
