## [Reviewer comments · BMJ Open]

ARTICLE DETAILS

TITLE (PROVISIONAL)	Implementation barriers and facilitators of remote monitoring, remote consultation and digital care platforms through the eyes of healthcare professionals: a review of reviews
AUTHORS	Oudbier, Susan; Souget-Ruff, S.P.; Chen, B.S.J.; Ziesemer, K.A.; Meij, J.J.; Smets, Ellen M. A.

VERSION 1 – REVIEW

REVIEWER	Krzesiński, Paweł Military Institute of Medicine - National Research Institute, Department of Cardiology and Internal Diseases
REVIEW RETURNED	08-Jun-2023

GENERAL COMMENTS	Dear Authors The paper is well written and concerns a very important topic. Congratulations I have only one minor comment regarding Table 4: What does "a" in super script mean? There is no red as I see, is it correct? Please state clearly in the legend what the numbers in round brackets means (I suppose references).
---

REVIEWER	Sanchez-Vazquez, Antonio Anglia Ruskin University, Faculty of Medical Science
REVIEW RETURNED	21-Jun-2023

GENERAL COMMENTS	Dear Authors, I think the article is well written and addresses the important area of the use of technology as a support tool for the delivery of healthcare. For some time technology has not been living up to its potential and this can expand on that theme and contribute to the current conversation regarding digital transformation in healthcare, especially from the HCP perspective. Overall while the findings are interesting, can they be brought out a bit more clearly in the conclusions, maybe via a diagrammatic representation, or something similar. Can you bring out more in the summary the original perspective of this paper. When they say 'technology implementation' what does that mean and can it be positioned a bit more clearly as to the 'digital transformation' of what: of work? of healthcare? of patient management, patient experience, health outcomes, etc...
---

	Otherwise I have a few comments for your consideration, more specifically:  1. The review seems quite systematic and followed Prisma guidelines, should it not be entitled as such? 2. In 'Information sources and search' section you refer to 3 articles that could not be found. Could the British Library be of help? 3. In the 'Review characteristics' section would it be worth including Mental Health (n=4), given the growing awareness and importance of this area of health and wellbeing. 4. On Table 3, it may be worth specifying that '-' is a barrier, and '+' a facilitator? 5. Also in Table 3 it may be clearer if the % of Reviews contained the numerical values as well as the bar chart? 6. In the Discussion section (P20/Line 48-49/ What does it mean by common organisational barriers?) This seems an important section, can this paragraph be reinforced? 7. Also in the Discussion section (P21/end of L17-26). It seems to be an important finding, if patients are satisfied with remote monitoring or telehealth, even if the ECP perceiving these as barriers, what does this mean for aligning best care with best practice? 8. There seem some important findings in the Conclusions section. Focus on the Organisational perspective, the patient-HCP perspective, the financial/resources perspective, the mechanics of training, and the importance of co-creation involving patients and HCP from inception (how could this be accomplished, given most apps and devices are manufactured by third party companies). Can the implications for future practice be better linked to the text in this section. Perhaps with a diagram?
--	---

VERSION 1 – AUTHOR RESPONSE

Reviewer 1

Comments to the Author:

1)The paper is well written and concerns a very important topic. Congratulations

Our response: Thank you for your positive feedback.

2) I have only one minor comment regarding Table 4: What does "a" in super script mean?

There is no red as I see, is it correct? Please state clearly in the legend what the numbers in round brackets means (I suppose references).

Our response: it is true that there is no red background, which means that there are no references that only mentioned barriers. To avoid confusion, we have removed this.

We have now added an explanation on the brackets in table 4.

Changes to the manuscript:

Results, Table 4, page 21: removed; red background means reported as barrier.

Results, Table 4, page 21: ^aThe numbers in parentheses denote the corresponding references.

Reviewer 2

Comments to the Author:

I think the article is well written and addresses the important area of the use of technology as a support tool for the delivery of healthcare. For some time technology has not been living up to its potential and this can expand on that theme and contribute to the current conversation regarding digital transformation in healthcare, especially from the HCP perspective.

Our response: we thank the reviewer for the kind words on our systematic review.

Changes to the manuscript: None.

Overall while the findings are interesting, can they be brought out a bit more clearly in the conclusions, maybe via a diagrammatic representation, or something similar.

Our response: We agree with the reviewer that the conclusions can be presented more clearly. Therefore, we added an extra diagram to the review.

Changes to the manuscript: Discussion, page 25:

Can you bring out more in the summary the original perspective of this paper. When they say 'technology implementation' what does that mean and can it be positioned a bit more clearly as to the 'digital transformation' of what: of work? of healthcare? of patient management, patient experience, health outcomes, etc...

Our response: We appreciate the reviewer's suggestion regarding the need for consistency in terminology throughout the abstract. In response, we have carefully revised the manuscript to address this concern by making a number of changes regarding the terminology.

Changes to the manuscript: Abstract, page 2:

Digital transformation in healthcare

Digital technologies → digital health technologies

Introduction, page 6: Digital health – as defined by the world health organisation

Otherwise I have a few comments for your consideration, more specifically:

1. The review seems quite systematic and followed Prisma guidelines, should it not be entitled as such?

Our response: Thank you for pointing out that we can make more clear that we followed a systematic process, guided by the Prisma criteria. We added this to the abstract, and to the strengths of this review in the bullet points.

Changes to the manuscript: Abstract, page 2: Systematic review of reviews following the Preferred Reporting Items for Systematic Reviews and Meta-Analyses (PRISMA) statement.

Strengths and limitations of this study, page 4: This systematic review encompassed digital health technologies which include patient interaction

2. In 'Information sources and search' section you refer to 3 articles that could not be found. Could the British Library be of help?

Our response: We appreciate the reviewer's suggestion. We would like to assure that we already took great care in addressing this concern, by consulting our librarian in the search process to retrieve any missing articles. This resulted in only these three articles remaining unavailable.

Changes to the manuscript: None.

3. In the 'Review characteristics' section would it be worth including Mental Health (n=4), given the growing awareness and importance of this area of health and wellbeing.

Our response: We already included a category called mental health problems. We agree with the reviewer to consolidate this into mental health.

Changes to the manuscript: Results, review characteristics, page 13: The majority of digital technologies examined in all reviews encompassed telehealth or remote monitoring of patients (Table 1). Most of the diseases examined are cardiovascular disease (n=9), mental health (n=7), diabetes mellitus (DM) (n=5), and chronic obstructive pulmonary disease (n=4).

4. On Table 3, it may be worth specifying that '-' is a barrier, and '+' a facilitator?

Our response: thank you for pointing this out. We have added this to Table 3.

Changes to the manuscript: Results, Table 3, page 19: Barrier (-) or facilitator (+)

5. Also in Table 3 it may be clearer if the % of Reviews contained the numerical values as well as the bar chart?

Our response: we thank the reviewer for this feedback and added the numerical values to Table 3.

Changes to the manuscript: Results, Table 3, page 19: #of reviews

6. In the Discussion section (P20/Line 48-49/ What does it mean by common organisational barriers?) This seems an important section, can this paragraph be reinforced?

Our response: thank you for pointing this out. We added an explanation to this section.

Changes to the manuscript: Discussion, page 22: This review of reviews demonstrates that the majority of research in the past years on barriers and facilitators addressed common organisational barriers of technology implementation, such as lack of training, lack of time and resources, an increase in workload, and lack of leadership and support. This substantial attention to organisational issues can be interpreted in two ways. Firstly, organisational barriers might remain prevalent for implementation, hence the substantial literature on them. Alternatively, it suggests that organisational aspects could dominate discussions about implementation of technology, potentially overshadowing other levels on which barriers can take place. For instance...

7. Also in the Discussion section (P21/end of L17-26). It seems to be an important finding, if patients are satisfied with remote monitoring or telehealth, even if the HCP perceiving these as barriers, what does this mean for aligning best care with best practice?

Our response: We thank the reviewer for raising this interesting point regarding a conflict between the needs of the patient versus the needs of health care professional. We have addressed this more explicitly in the review.

Changes to the manuscript: Discussion, page 23: In this regard, there is a discrepancy between the needs and perspectives of HCPs and patients. The most used explanation on differences in satisfaction levels with regard to videoconsultation is the inability of HCPs to physically examine the patient (66). While this review primarily focuses on the perspective of HCP, patients are also significant end users of technology. Therefore, it is essential to consider and identify the varying needs of patients in addition to those of HCPs, by conducting more research on the patients' perspectives on the influence of digital technology on their care, and the HCP-patient relationship specifically. Such research may yield more discrepancies, which need to be solved to provide patients with best care without unduly burdening HCP.

8. There seem some important findings in the Conclusions section. Focus on the Organisational perspective, the patient-HCP perspective, the financial/resources perspective, the mechanics of training, and the importance of co-creation involving patients and HCP from inception (how could this be accomplished, given most apps and devices are manufactured by third party companies). Can the implications for future practice be better linked to the text in this section. Perhaps with a diagram?

Our response: we thank the reviewer for pointing this out. We tried to link the identified barriers and facilitators from the perspective of the HCP to implications for practice by including the figure to the review.

Changes to the manuscript: Discussion, page 25

VERSION 2 – REVIEW

REVIEWER	Sanchez-Vazquez, Antonio Anglia Ruskin University, Faculty of Medical Science
REVIEW RETURNED	07-Apr-2024

GENERAL COMMENTS	Dear Authors, Thank-you for your revisions. Overall I think you have addressed most of the issues and have reinforced the article in that respect. That said, I have a few more comments, which it may be worth considering: 1. My comment about bringing out more the original perspective of the paper, in terms of more clearly presenting the transformation (or what) has not really been addressed. This was not a point of terminology, but of clarity in the narrative (see original comment). The addition of 'health' and 'healthcare' in two places do not really provide this clarification.2. The new diagram (which is in the responses letter) provides a much clearer summary of the conclusions, but it does not seem to be included in the tracked changes version of the paper I have, nor is there any new text to support the inclusion of the diagram. Am I missing something?3. Finally, I do agree with the editor's comments that the search could be updated to include more current articles. These are only from last 2 years current years, so should not prove too laborious. If not maybe a rationale can be provided as to why only papers until 2022 (now 2 years ago), and if not maybe this date can be included in the title (...a review of reviews published before June 2022")
---

VERSION 2 – AUTHOR RESPONSE

Reviewer's comments

My comment about bringing out more the original perspective of the paper, in terms of more clearly presenting the transformation (or what) has not really been addressed. This was not a point of terminology, but of clarity in the narrative (see original comment). The addition of 'health' and 'healthcare' in two places do not really provide this clarification.

Our response: We agree with the reviewer that the narrative on digital transformation could be more clearly presented in the beginning of the paper. Therefore, we added some extra sentences to the introduction.

Changes to the manuscript: Introduction, page 5: Globally, healthcare faces huge challenges due to an increase in individuals living with chronic diseases, an aging population, and increasing healthcare expenditures (1). Therefore, a so called digital transformation of healthcare is advocated considering the opportunities the implementation of technology has in terms of reducing costs (2), improving quality of care (3), and reducing the burden on healthcare professionals (HCPs) (4).

The new diagram (which is in the responses letter) provides a much clearer summary of the conclusions, but it does not seem to be included in the tracked changes version of the paper I have, nor is there any new text to support the inclusion of the diagram. Am I missing something?

Our response: Due to journal regulations, figures must be extracted from the manuscript and included in the supplementary material. Therefore, the figure is not included in the original manuscript.

Changes to the manuscript: Discussion, page 19: One way to overcome organisational barriers is by addressing them in the implementation strategy (Figure 2).

Figure legend, page 32: Figure 2 - Top three barriers or facilitators on each level of digital health implementation, and practical implications or solutions.

Finally, I do agree with the editor's comments that the search could be updated to include more current articles. These are only from last 2 years current years, so should not prove too laborious. If not maybe a rationale can be provided as to why only papers until 2022 (now 2 years ago), and if not maybe this date can be included in the title (...a review of reviews published before June 2022")

Our response: We understand the reviewer's wish for an update of the search. Unfortunately, as argued when submitting our revision, due to time constraints exacerbated by the prolonged duration of the review process, we have opted not to undertake the search update at this juncture. To acknowledge the concern raised by the reviewer, we have now included this as a limitation in our discussion. We sincerely hope for your understanding in this regard.

Changes to the manuscript: Discussion, strengths and limitations, page 22

However, the current review was limited by digital technology and policies regulating these being in a state of constant flux. Therefore, barriers and facilitators reported in older studies might be less relevant over time. Presently, we acknowledge that the current state of affairs is undergoing rapid evolution, and consequently, the data of our review may not be entirely up-to-date. However, given that half of the reviews included are less than five years old, we do not expect more recent studies to have major impact on our findings.

VERSION 3 – REVIEW

REVIEWER	Sanchez-Vazquez, Antonio Anglia Ruskin University, Faculty of Medical Science
REVIEW RETURNED	08-May-2024
GENERAL COMMENTS	Dear Authors, Thank-you for your revisions. From my perspective these are sufficient to address the issues raised in my comments from April 2024. I am happy to now recommend it for publication.